# Application of Second Law Analysis in Heat Exchanger Systems

**DOI:** 10.3390/e21060606

**Published:** 2019-06-19

**Authors:** Seyed Ali Ashrafizadeh

**Affiliations:** Department of Chemical Engineering, Dezful Branch, Islamic Azad University, Dezful 6461645169, Iran; ashrafi@iaud.ac.ir; Tel.: +98-61-4225-4683

**Keywords:** heat exchangers, second law analysis, entropy generation minimization

## Abstract

In recent decades, the second law of thermodynamics has been commonly applied in analyzing heat exchangers. Many researchers believe that the minimization of entropy generation or exergy losses can be considered as an objective function in designing heat exchangers. Some other researchers, however, not only reject the entropy generation minimization (EGM) philosophy, but also believe that entropy generation maximization is a real objective function in designing heat exchangers. Using driving forces and irreversibility relations, this study sought to get these two views closer to each other. Exergy loss relations were developed by sink–source modeling along the heat exchangers. In this case, two types of heat exchangers are introduced, known as “process” and “utility” heat exchangers. In order to propose an appropriate procedure, exergy losses were examined based on variables and degrees of freedom, and they were different in each category. The results showed that “EGM” philosophy could be applied only to utility heat exchangers. A mathematical model was also developed to calculate exergy losses and investigate the effects of various parameters. Moreover, the validity of the model was evaluated by some experimental data using a double-pipe heat exchanger. Both the process and utility heat exchangers were simulated during the experiments. After verifying the model, some case studies were conducted. The final results indicated that there was not a real minimum point for exergy losses (or entropy generation) with respect to the operational variables. However, a logic minimum point could be found for utility heat exchangers with regard to the constraints.

## 1. Introduction

From energy and environmental viewpoints, the application of a method in order to perform highly efficient heat transfer is of the essence. Thermodynamic laws provide designers and engineers with a powerful and efficient tool known as exergy analysis. Moreover, energy balance can provide useful information about the status of a system as far as plant body losses and incomplete combustion are concerned. No difference, however, has been found between energy types, and losses have not been considered due to a decrease in energy quality. By definition, exergy is referred to as the maximum shaft or electrical work in a reversible process when the system reaches environmental conditions. Exergy analysis makes it possible to provide thermodynamic development. Economic and environmental considerations, though, determine the selection of the final scenario. Contrary to what comes true with energy, exergy vanishes as a result of irreversibilities. In other words, any kind of irreversibility leads to exergy losses. A major part of the literature has described such relevant topics. Bejan [1,2], for example, has linked the principles of heat transfer to the second law of thermodynamics and entropy generation. Szargut et al. [3], Kotas [4], Dincer and Rosen [5], and Goran [6] have all delved further into developing and applying the exergy analysis concept through various processes. Sciubba and Zullo [7] have addressed problems with exergy existence and quantification in non-equilibrium systems. Lucia [8] has analyzed the exergy and entropy generation of open systems while taking environmental effects into consideration. Lucia and Grisolia [9] have suggested the second law approach for anticancer therapies. Arto Annila [10] has proposed a relationship between irreversibility and diminishing energy density variations during the shortest time according to the second law of thermodynamics.

Today, heat exchangers are widely used in a variety of industries. Designing a heat exchanger involves the consideration of both rates of heat transfer between fluids and mechanical power expanding to overcome the fluid friction and movement of the fluids by the heat exchanger. An analysis of the second law allows heat exchanger designers to consider both factors simultaneously: However, this is not possible using first law analysis. A systematic design of heat exchangers using second-law-based procedure is thus required. Many works have focused on this subject in the literature. Clintock [11] carried out one of the first studies in this field through applying the irreversibility concept to designing a heat exchanger. Bejan [12,13] explained two contributions to exergy loss (namely through heat transfer across finite temperature difference and through fluid friction in channels). He presented a broad spectrum of design optimization for simple co-current and countercurrent heat exchangers. Sarangi and Choowdhury [14] analyzed entropy generation in a counterflow heat exchanger and derived expressions in terms of relevant dimensionless parameters. Sekulic [15,16], investigated the entropy generation in different types of heat exchangers and examined the effect of parameters such as the inlet temperature ratio and fluid flow heat capacity ratio on the quality of energy transformation. Nag and Mukherjee [17] derived expressions for the entropy generation rate in a connective heat transfer using the constant wall temperature. Grazzini and Gori [18] carried out a study according to the number of entropy generations in order to optimize heat exchanger designs for counter heat exchangers. Das and Roetzel [19,20] presented an exergetic analysis of a plate heat exchanger with the axial dispersion in fluid through including flow misdistribution and back mixing. They also reported the various natures of different contributions to the total exergy loss in the heat exchanger with respect to axial dispersion parameters of the Peclet number and the propagation velocity of the dispersion wave. Ogulata et al. [21] analyzed the behavior of parameters such as optimum flow path length, dimensionless mass velocity, and dimensionless heat transfer zone for a crossflow plate-type heat exchanger operating through unmixed fluids. Mahmud and Fraser [22,23,24] analyzed the second law for heat exchangers using cylindrical annular space. They analyzed the heat transfer and fluid flow inside the channel using two parallel plates and examined forced convection in a circular duct for non-Newtonian fluids. Shah and Sekulic [25] thoroughly explained the necessity of considering both the first and second laws in designing heat exchangers. In another study, they [26] discussed the relationship between heat exchanger effectiveness and temperature difference irreversibility for complex flow arrangements. Naphon [27] analyzed the second law for horizontal concentric tubes in heat exchangers. Gupta and Das [28] also conducted an analysis on crossflow heat exchangers for non-uniform flow. Inspired by the minimum entropy production principle [29], Bejan developed a entropy generation minimization (EGM) approach in order to optimize heat exchanger designs. He also demonstrated that EGM may be used by itself in the preliminary design stages to detect trends and the existence of optimization opportunities. Other researchers have applied EGM philosophy to optimize some aspect of heat exchangers, including configuration [30], porous medium [31], fins and pins [32], and uniform wall temperature [33].

In contrast, due to some other reasons, a lot of research has not only rejected entropy generation minimization but also has focused on entropy generation maximization as an objective function in designing heat exchangers. An example of such studies is Fakheri’s work [34], as he has claimed that the philosophy of entropy generation minimization is based on three assumptions:It is possible to minimize entropy production (exergy destruction) in a heat exchanger;It is desirable to minimize entropy production (exergy destruction) in a heat exchanger;Similar to isentropic efficiency, heat exchanger effectiveness is associated with irreversibility.

Moreover, he considered these three assumptions to be unacceptable and, in fact, assumed that the entropy minimization approach is not an appropriate tool for analyzing heat exchangers and that it should not have been used as an objective in designing heat exchangers.

Using the relationship between driving forces and irreversibility, this paper mainly aims to make the views of the “EGM” proponents and opponents closer to each other. Driving forces are needed to perform real phenomena. They, along with the transfer coefficient, affect the heat transfer rate. On the other hand, with regard to the relationship between these forces and irreversibility, they influence exergy losses. Driving forces can thus be used as a relationship between the heat transfer rate, transfer coefficients, and exergy losses. In the present study, an analysis of the second law was run based on this relation in heat exchangers and exergy losses as a function of driving forces. A heat exchanger’s exergy losses were evaluated in various process conditions. This evaluation indicated the places where “EGM” can be efficiently applied.

## 2. Sink–Source Model Exergy Analyses in Heat Transfer Processes

In all heat transfer processes performed above the environment’s temperature, higher and lower temperature heat resources can be considered to be an exergy source and sink, respectively. According to the second law of thermodynamics, an exergy sink cannot absorb source exergy as a whole. Thus, in the sink–source model, exergetic efficiency and exergy losses are defined as follows:(1)$$\eta =\frac{\Delta EX_{sink}}{\Delta EX_{source}},$$
(2)$$EL=\Delta EX_{source}-\Delta EX_{sink},$$
where *η* is the exergetic efficiency, *EL* is exergy loss, and Δ*EX_source_* and Δ*EX_sink_* are the source and sink exergy changes, respectively. As shown in Equations (1) and (2), if the exergy sink can absorb higher amounts of exergy from the source, exergy losses and exergy efficiency become lower and higher, respectively. Exergy losses in convection heat transfer systems are divided into two components, one associated with temperature differences and the other one with fluid pressure drops. In the first component (if the system temperature is above the environment temperature), the part of the system with the higher temperature is the source, while the other part is the sink. In the second component, the instrument that moves the fluids (pump, compressor, etc.) is the source, and the pressure change of the fluids is the exergy sink.

### 2.1. Basic Relations

Exergy is the maximum amount of obtainable work or power when the system is brought from its initial state to a thermodynamic equilibrium state with the common substance of the natural environment using reversible processes, which only involves an interaction with the above-mentioned substances of nature. In other words, exergy is the maximum amount of work that can be obtained if a substance or a form of energy is converted to its inert reference state [35]. According to the first and second laws of thermodynamics,
(3)$$W_{S}=\Delta H-Q.$$

In Equation (3), it is assumed that kinetic and potential energy variations are negligible. Equation (4) is
(4)$$EX=W_{max}=\left(H-H_{0}\right)-T_{0}\left(S-S_{0}\right).$$

In Equation (4), the term (*EX*) on the left hand is exergy, and the subscript “0” denotes environment conditions. The differential form of Equation (4) results in
(5)$$dEX=dh-T_{0}ds.$$

According to the definition of enthalpy and entropy,
(6)$$dEX=c_{P}dT-T_{0}\left(\frac{du+dw}{T}\right)=c_{P}dT-\frac{T_{0}}{T}\left(dh-vdP\right).$$

In Equation (6), ν is specific volume. After rearranging the relations in Equation (6), the following equation can be presented:(7)$$dEX=c_{P}\left(1-\frac{T_{0}}{T}\right)dT+\left(\frac{T_{0}}{T}\right)vdP.$$

The degree of dependence of exergy on temperature and pressure can be calculated by Equation (7):(8)$${\left(\frac{\partial EX}{\partial T}\right)}_{P}=c_{P}\left(1-\frac{T_{0}}{T}\right),$$
(9)$${\left(\frac{\partial EX}{\partial T}\right)}_{T}=T_{0}\left(\frac{v}{T}\right).$$

Exergy variations of Equations (7)–(9) are as follows:(10)$$\Delta EX=\int_{T_{1}}^{T_{2}}c_{P}\left(1-\frac{T_{0}}{T}\right)dT+T_{0}\int_{P_{1}}^{P_{2}}\frac{v}{T}dP$$
or
(11)$$\Delta EX=c_{Pm}\left(T_{2}-T_{1}\right)-c_{Pm}T_{0}Ln\left(\frac{T_{2}}{T_{1}}\right)+T_{0}\int_{P_{1}}^{P_{2}}\frac{v}{T}dP.$$

There are two sources of irreversibilities in heat exchangers, (1) a heat transfer and (2) fluid friction. Hence, there are two kinds of exergy losses: (1) Exergy losses resulting from irreversibilities from the heat transfer ($\Delta EX_{\Delta T}$) and (2) exergy losses resulting from irreversibilities of the fluid flow ($\Delta EX_{\Delta p}$). The two exergy losses can be written as Equation (11). Equation (12) is
(12)$$\Delta EX_{\Delta T}=c_{Pm}\left(T_{2}-T_{1}\right)-c_{Pm}T_{0}Ln\left(\frac{T_{2}}{T_{1}}\right)=\Delta H\left(1-\frac{T_{0}}{T_{lm1,2}}\right),$$
(13)$$\Delta EX_{\Delta P}=T_{0}\int_{P_{1}}^{P_{2}}\frac{v}{T}dP.$$

In Equation (12), *T*${}_{LM1,2}$ is the logarithm mean temperature between initial and final fluid states. For total exergy losses, we then have
(14)$$\Delta EX_{total}=\Delta EX_{\Delta T}+\Delta EX_{\Delta P}.$$

In Equation (13), fluid state equations are needed. If the fluids are assumed to be an ideal gas, then
(15)$$\Delta EX_{\Delta P^{ig}}=nRT_{0}ln\left(\frac{P_{2}}{P_{1}}\right).$$

It is proven that Equation (15) is valid for real gases [4].

The compressible fluid functionality of (ν/T) with respect to the pressure effect is, however, negligible:(16)$$\Delta EX_{\Delta P^{liquid}}≅\;T_{0}\left(\frac{v}{T}\right)\Delta P.$$

Consequently, exergy loss calculations can be as follows:(17)$$\Delta EX_{gas}=\Delta H\left(1-\frac{T_{0}}{T_{lm1,2}}\right)+nRT_{0}Ln\left(\frac{P_{2}}{P_{1}}\right),$$
(18)$$\Delta EX_{liquid}=\;\Delta H\left(1-\frac{T_{0}}{T_{lm1,2}}\right)+T_{0}\left(\frac{v}{T}\right)\Delta P.$$

It should be noted that the exergy losses induced by the pressure drop are negligible in liquids.

### 2.2. Calculation of Exergy Losses in Heat Exchangers

As was previously pointed out, total exergy variations in heat exchangers include $\Delta EX_{\Delta T}$ and $\Delta EX_{\Delta p}$. For total exergy losses, the following equation can then be written
(19)$$EL_{total}=EL_{\Delta T}+EL_{\Delta P}.$$

In Equation (19), *EL* is total exergy loss and $EL_{\Delta T}$ and $EL_{\Delta P}$ are exergy losses arising from the temperature and pressure differences, respectively (heat and flow driving forces). The sink–source model can be used for the $EL_{\Delta T}$ calculation. Consider a heat exchanger above the environment’s temperature:(20)$$EL_{\Delta T}=\Delta EX_{\Delta Thot}-\Delta EX_{\Delta Tcold}.$$

In Equation (20), $\Delta EX_{\Delta Thot}$ and $\Delta EX_{\Delta Tcold}$ are exergy variations in hot and cold fluids, respectively. By applying Equation (12) to Equation (20), the following equation is achieved:(21)$$EL_{\Delta T}=\Delta H_{hot}\left(1-\frac{T_{0}}{T_{LMH}}\right)-\Delta H_{cold}\left(1-\frac{T_{0}}{T_{LMC}}\right).$$

According to the first law of thermodynamics, $\Delta H_{hot}$ and $\Delta H_{cold}$ are equal to the rate of heat transfer (*q*): Hence, the exergy losses resulting from the heat transfer can be calculated by Equation (22):(22)$$EL_{\Delta T}=qT_{0}\left(\frac{T_{LMH}-T_{LMC}}{T_{LMH}\times T_{LMC}}\right).$$

In order to calculate the fluid flow exergy losses, there is no exergy interchange between fluids. Fluid transition systems (for hot and cold streams) work separately, and therefore
(23)$$EL_{\Delta P}=\Delta EX_{\Delta Phot}+\Delta EX_{\Delta Pcold}.$$

In Equation (23), $EL_{\Delta P}$ is the total exergy loss caused by fluid friction, and $\Delta EX_{\Delta Phot}$ and $\Delta EX_{\Delta Pcold}$ are hot and cold fluid exergy variations, respectively, which can be calculated by Equation (16).

### 2.3. The Second Law Analysis of Heat Exchangers

From a mechanical viewpoint, there are several types of heat exchangers. However, they can be operationally divided into two main groups:Process heat exchangers, where hot and cold fluids are both process fluids;Utility heat exchangers, where one of the fluids (hot or cold) comes from a utility unit and another from a process fluid.

In process heat exchangers, designers cannot freely change the operation parameters to evaluate their effects on entropy generation and exergy losses because the flow rates and input temperatures are inflicted on the designer beforehand. The output temperature of either fluid is also fixed in order to provide processing conditions in the following unit. If the output temperature for one of the fluids is fixed, according to the first law of thermodynamics, it is fixed for another one as well. In this case, only some mechanical parameters (e.g., size, weight, pressure drop, etc.) or economical parameters (e.g., cost) are thus disputable: However, irreversibilities and relevant parameters such as entropy generation or exergy losses are fixed and not controversial.

On the other hand, in utility heat exchangers, the designer can freely change one of the output or input temperatures or flow rate of the fluid provided from the utility unit and then investigate the effects of the second law analysis.

In order to have a better investigation, utility heat exchangers are divided into four groups:Liquid-cold utility heat exchangers;Liquid-hot utility heat exchangers;Gas-cold utility heat exchangers;Gas-hot utility heat exchangers.

#### 2.3.1. Liquid-Cold Utility Heat Exchangers

In this group, the fluid resulting from utility is a cooling liquid. Combining Equations (22) and (23), the following equation can be obtained:(24)$$EL_{total}=qT_{0}\left(\frac{T_{LMH}-T_{LMC}}{T_{LMH}\times T_{LMC}}\right)+\Delta EX_{\Delta Phot}.$$
As the cold fluid is liquid, the exergy losses caused by the fluid friction are neglected.

Thus we have Equation (25):(25)$$EL_{total}=\frac{qT_{0}}{T_{LMH}}\left(\frac{T_{LMH}\times Ln\left(\frac{T_{co}}{T_{ci}}\right)}{T_{co}-T_{ci}}-1\right)+\Delta EX_{\Delta Phot}$$
If the cold fluid input temperature is defined, the output temperature can be disputed.

Equation (25) is thus derived with respect to *Tco*:(26)$$\frac{dEL_{total}}{dT_{co}}=qT_{0}\left(\frac{\frac{1}{T_{co}}\left(T_{co}-T_{ci}\right)-Ln\left(\frac{T_{co}}{T_{ci}}\right)}{{\left(T_{co}-T_{ci}\right)}^{2}}\right)=0.$$

As can be observed in Equation (26), there is a maximum at *T_co_* = *T_ci_*. In other words, if there is a phase change in the cold fluid, the entropy generation or exergy losses are at a maximum.

If the input cold fluid temperature is considered to be a variable in Equation (25), the same results are obtained.

#### 2.3.2. Liquid-Hot Utility Heat Exchangers

In this kind of exchanger, the fluid coming from utility is a heating liquid. Combining Equations (22) and (23), the following equation can be obtained:(27)$$EL_{total}=qT_{0}\left(\frac{T_{LMH}-T_{LMC}}{T_{LMH}\times T_{LMC}}\right)+\Delta EX_{\Delta Pcold}.$$

As the hot fluid is liquid, the exergy losses caused by the fluid friction are neglected. If the hot fluid input temperature is defined, then the output temperature is disputable. Similarly to liquid-cold utility, this method leads to the same results: There is a maximum point for entropy generation or exergy losses when there is a phase change in the hot liquid.

#### 2.3.3. Gas-Cold Utility Heat Exchangers

In this kind of heat exchanger, the fluid resulting from utility is a cooling gas. Combining Equations (22) and (23), the following equation can be obtained:(28)$$EL_{total}=qT_{0}\left(\frac{T_{LMH}-T_{LMC}}{T_{LMH}\times T_{LMC}}\right)+\Delta EX_{\Delta Phot}+nRT_{0}Ln\left(\frac{P_{co}}{P_{ci}}\right).$$

As the heat transfer in heat exchangers is an isometric process, then
(29)$$\frac{P_{co}}{P_{ci}}\approx \frac{T_{co}}{T_{ci}}.$$

By inserting Equation (27) into Equation (26) and using the first law of thermodynamics, the following equation results:(30)$$EL_{total}=\left(qT_{0}+\frac{qRT_{0}}{M_{c}}\right)\frac{Ln\left(\frac{T_{co}}{T_{ci}}\right)}{\left(T_{co}-T_{ci}\right)}-\frac{qT_{0}}{T_{lmhot}}+\Delta EX_{\Delta Phot}.$$

All parameters except for the outlet cold fluid temperature are constant in Equation (30). It should be noted that the heat transfer rate is determined by the process fluid, i.e., the hot fluid. To investigate the exergy loss variation with respect to the outlet cold fluid temperature, the following equation can be presented:(31)$$\frac{dEL_{total}}{dT_{co}}=\left(qT_{0}+\frac{qRT_{0}}{M_{c}C_{c}}\right)\left(\frac{\frac{1}{T_{co}}\left(T_{co}-T_{ci}\right)-Ln\left(\frac{T_{co}}{T_{ci}}\right)}{{\left(T_{co}-T_{ci}\right)}^{2}}\;\right)=0.$$

The above equation is acceptable when the input and output cold temperatures are equal, indicating that the phase changes again. If the input cold temperature varies, the calculations lead to the same results.

#### 2.3.4. Gas-Hot Utility Heat Exchangers

In this kind of heat exchanger, the fluid that comes from utility is a heating gas. Combining Equations (22) and (23), the following equation can be obtained:(32)$$EL_{total}=qT_{0}\left(\frac{T_{LMH}-T_{LMC}}{T_{LMH}\times T_{LMC}}\right)+\Delta EX_{\Delta Phot},$$
(33)$$EL_{total}=\frac{qT_{0}}{T_{LMC}}-\left(qT_{0}+\frac{qRT_{0}}{M_{h}C_{h}}\right)\left(\frac{Ln\left(\frac{T_{ho}}{T_{hi}}\right)}{T_{ho}-T_{hi}}\right)+\Delta EX_{\Delta Pcold}.$$

A derivation from the above equation with respect to input or output hot fluid temperature leads to finding a maximum point for exergy losses at a constant temperature for this fluid (phase change).

As can be noticed, in all of these four exchangers, not only were the minimum exergy losses not found, but also the analysis revealed that condensers and evaporators widely used in industries are at a point with maximum exergy losses.

This leads to an illusion that indicates that entropy generation maximization is an objective in designing heat exchangers: However, the main reason for the wide application of condensers and evaporators is not to maximize exergy losses. In this case, the high heat transfer coefficient on a small scale and economic reasons are effective factors.

Another reason leading to this illusion is that an overall minimum point in the above analysis is impossible to achieve. To solve this problem, one should know that the inaccessibility of an ideal point does not justify an inclination toward the worst approach. Today, exergy losses and cost relations are a verity. These losses impose hidden costs that are notable on a large scale. Now, the posed question is as follows: What is the best operational point in heat exchangers? To answer this question, a computer code was developed using MATLAB software in order to investigate the effects of various operational parameters on exergy losses.

The model inputs are fluid properties, operational data, tube and shell diameter, flow kinds (co-current or countercurrent), and the fluid in the tube. After importing the data, the model calculates the required parameters, such as heat transfer rate, log mean temperature difference, the Reynolds number, heat transfer and friction coefficients, the heat exchanger length, the pressure drop, and finally exergy losses.

Designing heat exchangers is not the main objective of the model: However, its goal is to calculate exergy losses in different conditions. Only those parameters that are required for estimating exergy losses are thus calculated. The model accuracy was investigated using a pilot plant double-pipe heat exchanger. The next section contains an explanation of the experiments.

## 3. Experiments on Modeling Accuracy Investigation

### 3.1. The Applied System

In order to simulate an accuracy evaluation, some experiments were performed. A double-pipe pilot plant heat exchanger was used, as shown in Figure 1. The main parts of the pilot plant are shown in Figure 2.

The unit was set for testing and studying the behavior of double-pipe heat exchangers. This unit had a concentric double-pipe steel with an ability to change the type of flow to co-currents or countercurrents through changing the direction of the cold fluid flow. Cool water was supplied from a plumbing system. A tank equipped with a heater was used to heat water supplies. Hot water temperature was obtained from changing the heater’s power. Sensors were designed to monitor temperatures in different parts of the system.

The applied hot and cold fluids were water in this unit. Table 1 shows the detailed dimensions of this unit.

### 3.2. Experiments

The purpose of the experiments was to investigate the effect of parameters such as flow rates, temperature, and flow type (co-currents or countercurrents) on driving forces and exergy losses. Moreover, the model accuracy was also to be considered. In each experiment, the controlled parameters were the same inputs. The model and the uncontrolled parameters were the parameters calculated by the model, which were used for estimating the exergy losses. After each experiment, these parameters were employed to estimate exergy losses using a simulator.

According to the function of the two heat exchanger groups, some parameters were held constant and others changed freely. Co-current and countercurrent flows were investigated in all cases. Figure 3, Figure 4 and Figure 5 show the comparative results of the model and experiments for utility heat exchangers. The cooling water temperature was chosen to be similar to the cooling towers output.

Figure 6 and Figure 7 show the comparison of the experimental and model results in the process heat exchangers. The temperature of the hot water was determined with regard to the limits and power of the device.

In each experiment, controllable process conditions were selected as equal model inputs (controllable parameters). The other parameters required for calculating the exergy losses were the relaxed parameters given by the pilot plant (uncontrollable parameters). The experimental exergy losses were calculated based on these two parameter groups, indicated by cross marks in the above figures. The accuracy of the temperature thermocouples and flow meters were reported to be 0.1 and 0.01, respectively. The maximum computational error was 0.5: Hence, the uncertainty of the experimental data was equal to ±0.1006.

It can be noticed that the model had an acceptable consistency with the experiment results. In accordance with some pilot constraints, some case studies and the results are presented in the next section.

## 4. Results and Discussion

It was assumed that oil as a hot fluid is cooled by cooling the water coming from the utility unit. The flow rate and output temperature of the water were disputable, since water comes from the utility unit and is not a process fluid. The flow rate and input and output temperatures of the oil were 0.5 kg/s, 80 °C, and 35 °C, respectively. Figure 8 shows the heat exchanger exergy losses with respect to water outlet temperature.

As noticed, the exergy loss was reduced by increasing the water temperature: However, there were some other constraints from an operational perspective. The most important one was the water maximum temperature that must be returned to the utility unit. As this water must be cooled in the utility unit and reused, it cannot be above a certain amount. In the aforementioned example, the maximum allowable returned temperature for cooling water was assumed to be 40 °C: Hence, the minimum exergy loss was equal to 4170 W. Figure 9 represents this point graphically.

According to this figure, although there was no minimum point, a local minimum point could be found according to the constraints. Designing at this point could definitely reduce the costs imposed by the loss costs.

Now, 0.5 kg/s of oil was assumed to be heated by a hot water stream coming from the utility unit. In this case, the output hot temperature was disputable. Figure 10 shows the exergy losses against this temperature.

As can be observed, the exergy losses increased when the water output temperature rose. From the perspective of the second law, the heat exchanger’s efficiency was thus decreased by the rise in the water output temperature. From an operational viewpoint, however, the minimum water temperature returning to the utility unit was an important factor. As the water must be reheated, its temperature could not then be below a certain amount. In the above example, the minimum output hot water temperature coming back to the utility unit was assumed to be 60 °C: Hence, the minimum exergy loss was equal to 2290 W. Figure 11 shows this point graphically.

As can be noticed, a local minimum point could be found in accordance with the constraints. There is a procedure common in the two kinds of aforementioned heat exchangers, i.e., the direct relationship between exergy losses and log mean temperature differences. Figure 12 depicts this relationship.

Accordingly, the exergy losses were increased by raising the driving forces (temperature difference). On the other hand, from the viewpoint of the second law, the design must be performed with the lowest driving forces under the constraints in order to reach the local minimum point in exergy losses or entropy generation.

## 5. Conclusions

The exergetic behavior of heat exchangers was investigated using thermodynamics and heat transfer relations. Two kinds of heat exchangers (namely process and utility heat exchangers) were then introduced. A model was proposed to calculate the exergy losses and effects of various parameters. The accuracy of the model was evaluated using some experimental data. A case study on the inclusion of oil and water was also considered under two operational conditions, process and utility, after validating the model

The results showed that the “EGM” application in the utility heat exchangers was beneficial for the analysis and design, while the “EGM” could not be applied in the process heat exchangers.

Moreover, there was no real minimum point for exergy losses (or entropy generation) with respect to the process variables: However, a logic minimum point could be found according to the constraints for the utility heat exchangers. The type of heat exchanger (process or utility) must be identified before entropy generation analysis. The results of the second law of thermodynamics and entropy generation minimization can be applied to utility heat exchangers to optimize their operational conditions.

## Figures and Tables

**Figure 1 entropy-21-00606-f001:**
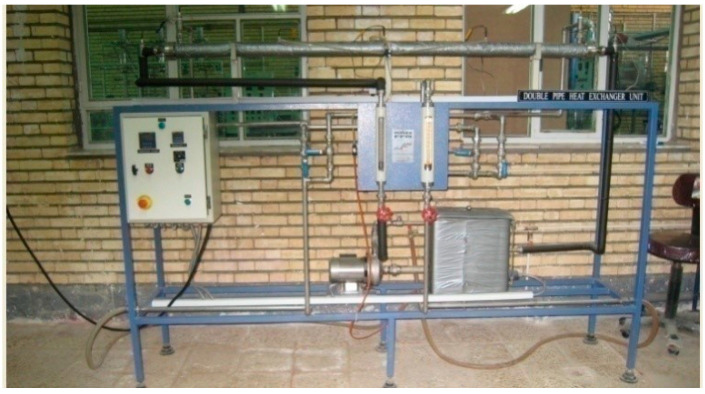
Applied system: Pilot plant double-pipe heat exchanger.

**Figure 2 entropy-21-00606-f002:**
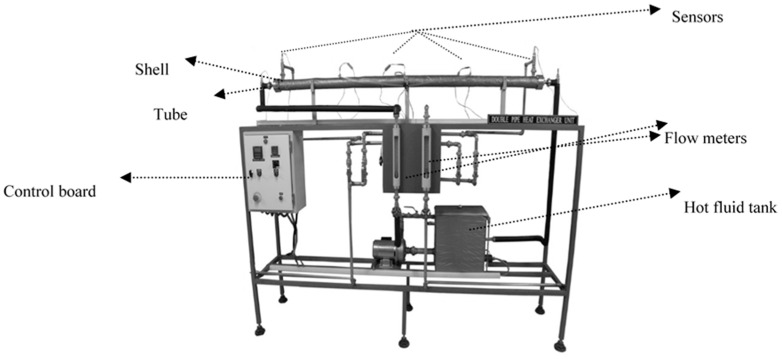
Main parts of the pilot plant.

**Figure 3 entropy-21-00606-f003:**
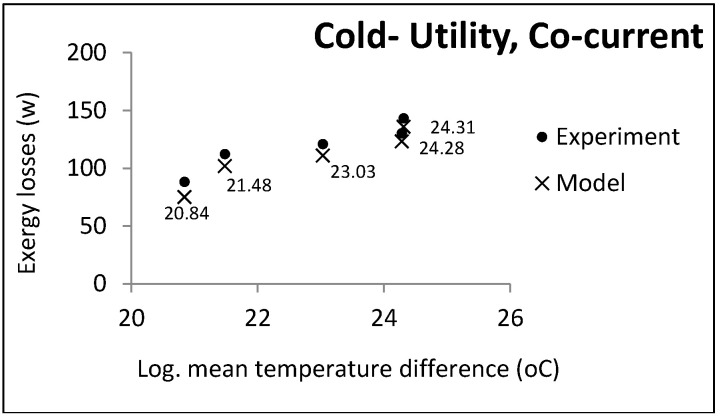
LMTD effect on exergy losses in utility heat exchangers (cold utility, co-current).

**Figure 4 entropy-21-00606-f004:**
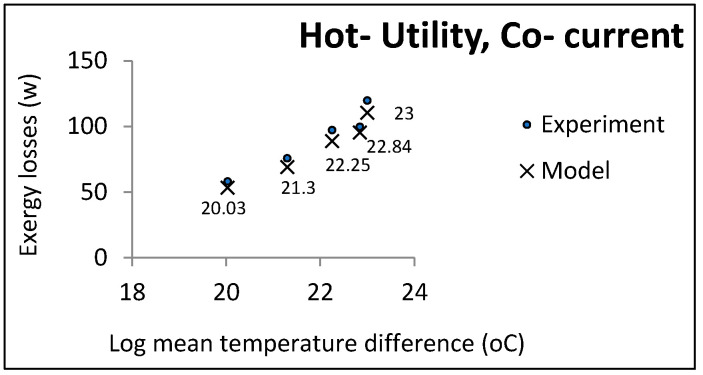
LMTD effect on exergy losses in utility heat exchangers (hot utility, co-current).

**Figure 5 entropy-21-00606-f005:**
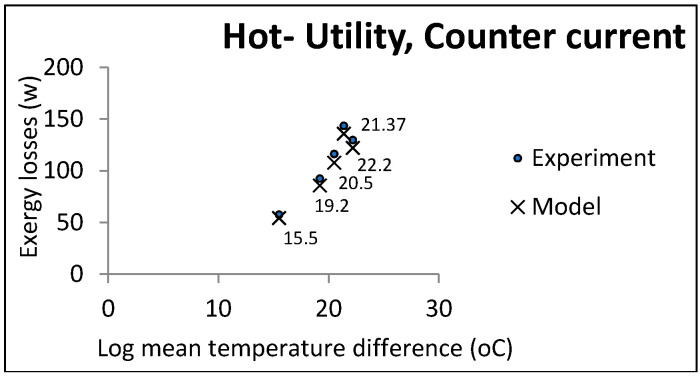
LMTD effect on exergy losses in utility heat exchangers (hot utility, countercurrent).

**Figure 6 entropy-21-00606-f006:**
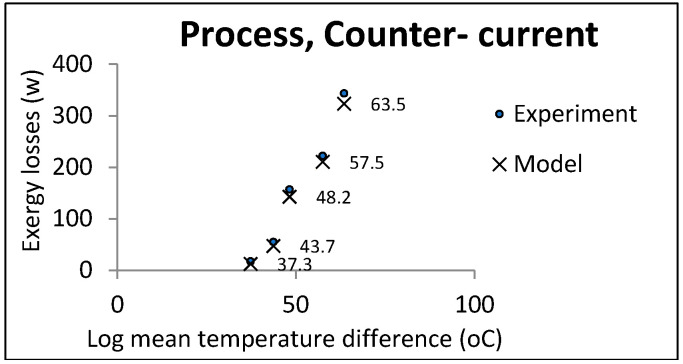
LMTD effect on exergy losses in process heat exchangers (countercurrent).

**Figure 7 entropy-21-00606-f007:**
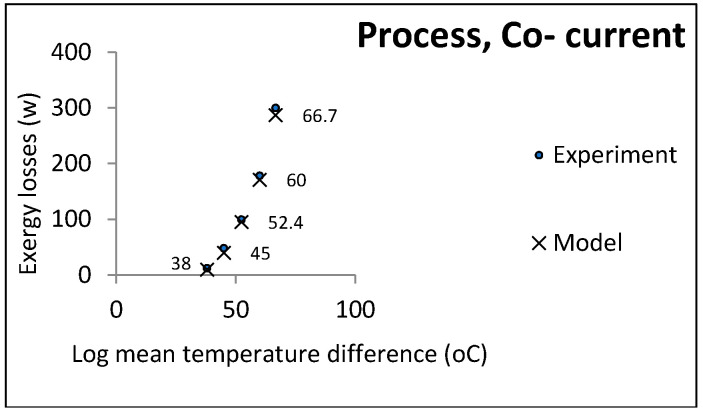
LMTD effect on exergy losses in process heat exchangers (co-current).

**Figure 8 entropy-21-00606-f008:**
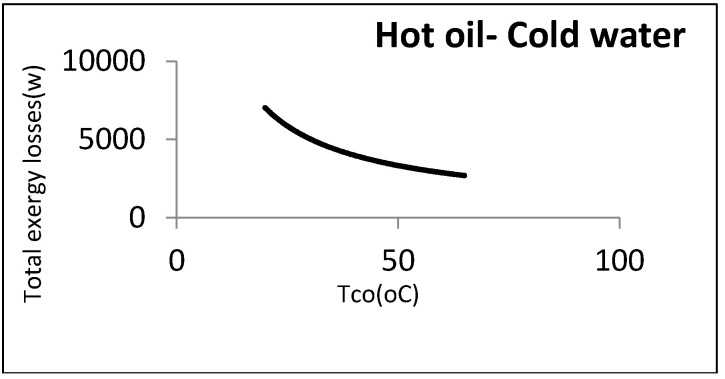
Exergy losses of heat exchanger with respect to cold water output temperature.

**Figure 9 entropy-21-00606-f009:**
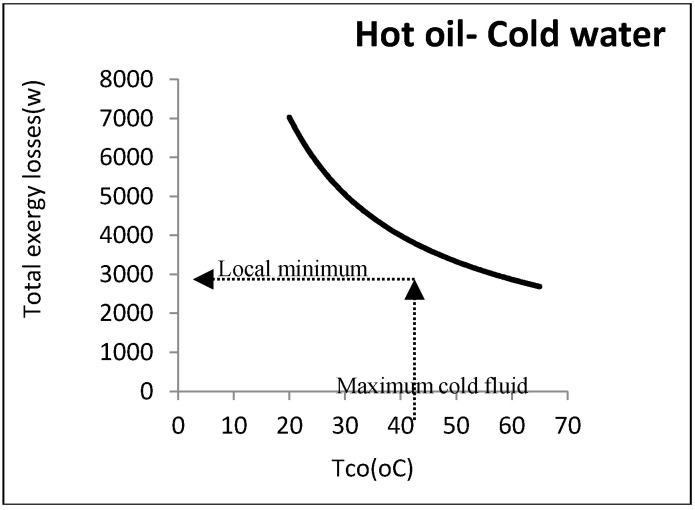
Local minimum exergy losses for cold-utility heat exchangers.

**Figure 10 entropy-21-00606-f010:**
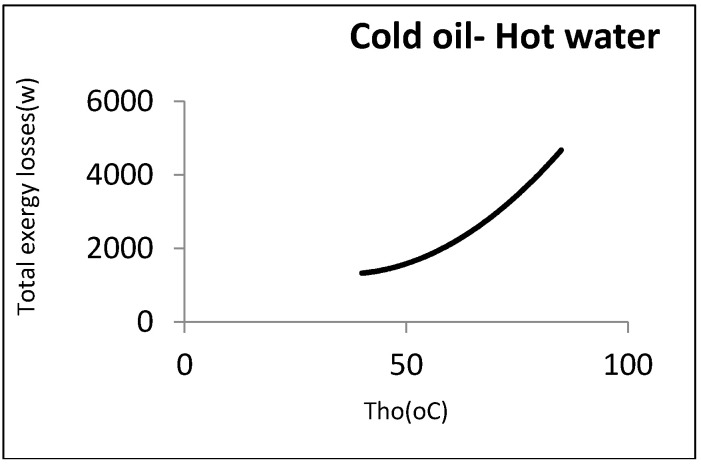
Heat exchanger exergy losses with respect to hot water outlet temperature.

**Figure 11 entropy-21-00606-f011:**
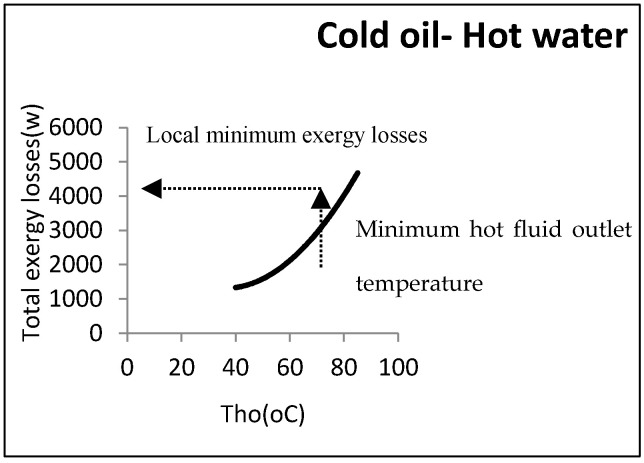
Local minimum exergy losses for hot-utility heat exchangers.

**Figure 12 entropy-21-00606-f012:**
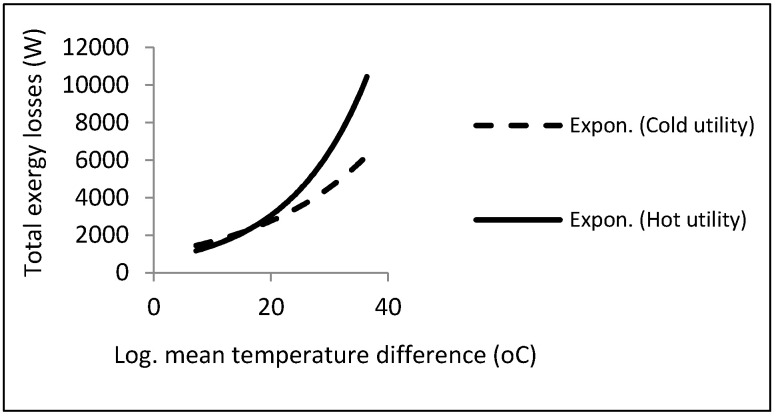
Effect of LMTD on exergy losses in cold- and hot-utility heat exchangers.

**Table 1 entropy-21-00606-t001:** The pilot plant dimensions (model: DZHT-05).

Tube Internal Diameter (mm)	Tube External Diameter (mm)	Shell External Diameter (mm)	Length (m)
21.3	36.2	42.4	2.5

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
