# Peer review of "Application of Second Law Analysis in Heat Exchanger Systems"

_entropy, 2019, doi:10.3390/e21060606_

Round 1

Reviewer 1 Report

The paper must be read carefully in order to correct some typos. Some are:

- line 67: [17, 18, & 19] --> [17, 18, 19]

- line 91: d4signing --> designing

- line 291: Kg/Sec --> kg/s

- line 301: 60^(oC) --> 60°C

Moreover, the paper could introduce some papers on the subject recently published. Some authors to be quoted are:

- Enrico Sciubba

- Umberto Lucia

- Giulia Grisolia

- Goran Wall

- Arto Annila

After these corrections the paper could be accepted.

Author Response

Dear Reviewer,

Thank you for the constructive comments which I have read in depth and breadth as you will note, they have been taken into consideration before submitting the revised draft. 

Comment 1:

The paper must be read carefully in order to correct some typos. Some are:

- line 67: [17, 18, & 19] --> [17, 18, 19]

- line 91: d4signing --> designing

- line 291: Kg/Sec --> kg/s

- line 301: 60^(oC) --> 60°C

Answer: The typo errors was corrected. Thank you very much.

Comment 2:

Moreover, the paper could introduce some papers on the subject recently published. Some authors to be quoted are:

- Enrico Sciubba

- Umberto Lucia

- Giulia Grisolia

- Goran Wall 

- Arto Annila

    Answer: Thank you for introducing this helpful references. These papers were carefully reviewed and added to the references.

Best Regards

S.A.Ashrafizadeh, Ph.D. Energy Eng.

Assistant Professor 
************************************* 
Chemical Engineering Department 
Engineering Faculty 
Islamic Azad University, Dezful branch 
Dezful-IRAN 
Tel:  +98-61-42420800 
Fax: +98-61-42420051 
[email protected][email protected]

Reviewer 2 Report

This is an interesting paper that is examining a topic that has become topical in recent years. Second law analysis of heat exchangers has been of interest for several reasons from a thermodynamic perspective but has not shown any practical application because, from my own experience of designing high performance heat exchangers for aircraft, the external constraints such as weight, volume, pumping power, cost are more important than the thermodynamic design providing the required duty can be met. This is shown in the conclusions where a local operating point is highlighted on the figures which represents the local conditions.

Heat exchangers are essentially entropy generators and the second law analysis can be used to develop the Effectiveness-NTU equations so the concept of the paper is valid. 

The introduction and literature review are very good but the conclusions are weak. It would help if the conditions relating to the  local operating points shown on the figures could be described accompanied by a reason for choosing these points. Also a comment of the utility of this method as a design tool would be of interest. 

The paper needs to be proof read as there are some errors with the English especially line 75. The version I received showed figures in mono chrome but the text refers to a red colour on a figure.       

Author Response

Dear Reviewer,

Thank you for the constructive comments which I have read in depth and breadth as you will note, they have been taken into consideration before submitting the revised draft. 

The introduction and literature review are very good but the conclusions are weak. It would help if the conditions relating to the  local operating points shown on the figures could be described accompanied by a reason for choosing these points. Also a comment of the utility of this method as a design tool would be of interest. 

Answer: Thank you for your comment. The local operating points have shown on the figures in the experiments section of the revised manuscript.

According to your advice, the reason for choosing the points has explained in the revised manuscript (“Cooling water temperature has chosen similar to the cooling towers outlet.” and  “The temperature of the hot water is determined according to the limits and power of the device.”).

As the final conclusion, it can be mentioned: “The results of the second law of thermodynamics and entropy generation minimization can be applied for utility heat exchangers to optimize their operational conditions.” which added to the conclusion section of the manuscript.

The paper needs to be proof read as there are some errors with the English especially line 75. The version I received showed figures in mono chrome but the text refers to a red colour on a figure. 

Answer:

Thank you very much. The English was reviewed and polished  so that the reader will understand it more easily. Also I have corrected all typographical and language errors.

Best Regards

S.A.Ashrafizadeh, Ph.D. Energy Eng.

Assistant Professor 
************************************* 
Chemical Engineering Department 
Engineering Faculty 
Islamic Azad University, Dezful branch 
Dezful-IRAN 
Tel:  +98-61-42420800 
Fax: +98-61-42420051 
[email protected][email protected]

Reviewer 3 Report

This paper deals with the analysis of entropy generation and exergy losses. This paper is interesting can be published after major revision taking into account the following comments:

1 – Introduction part should be improved with recently published papers in Entropy on the considered topic.

2 – Experimental setup should be described in detail.

3 – The uncertainty of the experimental data should be estimated in detail.

4 – There are some misprints and grammar mistakes in the text.

Author Response

Dear Reviewer,

Thank you for the constructive comments which I have read in depth and breadth as you will note, they have been taken into consideration before submitting the revised draft. 

Comment 1:

 Introduction part should be improved with recently published papers in Entropy on the considered topic.

Answer:

Thanks for your reminders. Five new references [6 - 10] have been added to the revised manuscript which two of them [7 and 10] are recently published papers in Entropy.

Comment 2:

 Experimental setup should be described in detail.

Answer:

The details of the set up were described after figure 2 in the revised manuscript.

Comment 3:

The uncertainty of the experimental data should be estimated in detail.

Answer:

Thank you very much for this important comment. The uncertainty of the experimental data is estimated before the result and discussion section  of the revised manuscript.

Comment 4:

There are some misprints and grammar mistakes in the text.

Answer:

Thank you very much. The English was reviewed and polished  so that the reader will understand it more easily. Also I have corrected all typographical and language errors.

Best Regards

S.A.Ashrafizadeh, Ph.D. Energy Eng.

Assistant Professor 
************************************* 
Chemical Engineering Department 
Engineering Faculty 
Islamic Azad University, Dezful branch 
Dezful-IRAN 
Tel:  +98-61-42420800 
Fax: +98-61-42420051 
[email protected][email protected]

Round 2

Reviewer 1 Report

I suggest to accept the paper in the present form.

Reviewer 3 Report

Authors addressed all comments raised by Reviewer. I consider that this paper can be accepted for publication.